# ReCLLaMA: A Reasoning-Centered LLM Agent for Medical Diagnosis

## Abstract

Large Language Models (LLMs) have demonstrated impressive capabilities in natural language understanding, yet their application to clinical diagnosis remains constrained by hallucinations, limited interpretability, and the absence of formal reasoning mechanisms. To address these limitations, we propose **ReCLLaMA**, a **Reasoning-Centered LLM Agent for Medical Diagnosis**, which integrates statistical language models with symbolic inference over structured medical knowledge. ReCLLaMA aligns free-text symptom descriptions with standardized ontologies using pretrained biomedical encoders and performs logical reasoning over heterogeneous knowledge graphs constructed from EHR and pharmacological data. To reconcile representational mismatches across sources, we introduce a statistical entity alignment module based on random forest classification. This enables the construction of a unified knowledge space in which ReCLLaMA applies both deductive and abductive reasoning to derive interpretable diagnostic pathways. Our framework advances the theoretical integration of subsymbolic and symbolic AI in clinical contexts, offering a principled approach to traceable, knowledge-grounded decision-making. Empirical results on real-world datasets validate its superiority over black-box LLMs and rule-based systems in both accuracy and explainability.

## 1 Introduction

Knowledge graphs (KGs) have become essential for organizing and reasoning over complex relational data across a variety of domains. In areas such as finance, education, and manufacturing, KGs enable the integration of heterogeneous information sources to support decision-making, anomaly detection, and process optimization Zhang et al. (2023); Mo et al. (2024); Du et al. (2022). In the biomedical domain, KGs offer a principled way to represent structured knowledge extracted from clinical records, biomedical literature, and curated ontologies, thereby supporting critical tasks such as disease diagnosis, treatment planning, and personalized care Abdulla et al. (2023); Wu et al. (2024); Bonner et al. (2022).

Despite their effectiveness, traditional KG-based diagnostic systems remain constrained by their reliance on manually curated ontologies and rule-based reasoning frameworks—such as SNOMED-CT and UMLS—which hampers their scalability and limits their ability to adapt to the variability of unstructured clinical narratives Chang & Mostafa (2021); Amos et al. (2020). In parallel, recent advances in large language models (LLMs) have introduced a complementary paradigm: the ability to extract and synthesize medical knowledge directly from free-text inputs. Models like GPT-4 and MedPaLM have achieved notable success in clinical summarization and medical question answering OpenAI (2023); Qian et al. (2024).

However, LLMs remain prone to hallucinations and brittle reasoning. Unlike human intelligence—which relies on an interdependent cycle of *abduction* (forming explanatory hypotheses), *deduction* (deriving testable consequences), and *induction* (revising hypotheses based on observations) Peirce (1934); Harman (1965); Douven (2011)—most LLMs lack mechanisms for genuine iterative rule discovery. Recent evaluations often isolate these reasoning stages or remove interaction with the environment, obscuring how rules are actually tested and refined in practice Wang et al. (2022); He et al. (2024). As a result, LLMs may generate fluent but ungrounded outputs, overgeneralize from

incomplete cues, or misrepresent causal relations, particularly under sparse clinical feedback Huang et al. (2023); Guo et al. (2024).

Building on this perspective, we introduce **ReCLLaMA**—a **Re**asoning-**C**entered **LLM A**gent for **M**edical Di**a**gnosis. The core innovation of ReCLLaMA is its explicit integration of abductive, deductive, and inductive reasoning over biomedical knowledge graphs, enabling diagnostic support that is accurate, interpretable, and grounded in transparent rule-based logic. By leveraging the natural language understanding capabilities of large language models, ReCLLaMA further mitigates hallucinations—a persistent limitation of LLMs in the clinical domain—by anchoring unstructured inputs to structured reasoning pathways.

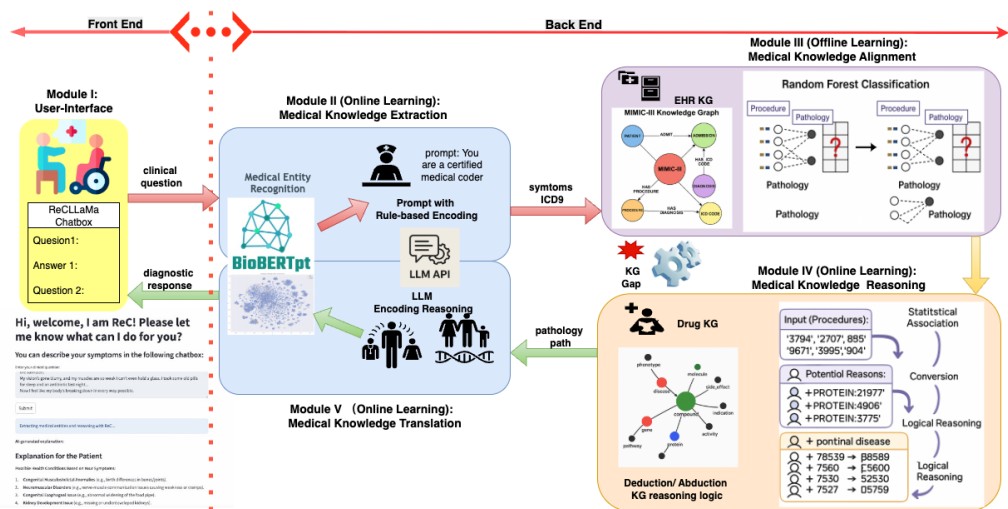

Figure 1: Overview of the ReCLLaMA framework. The system begins with a front-end chat interface that accepts free-text clinical questions or symptom descriptions from users. In **Module I**, these inputs are parsed through a medical chatbot interface. **Module II** performs medical knowledge extraction using BioBERT-based entity recognition and LLM-guided prompts to produce structured ICD-9 entities. These are aligned with the EHR-based MIMIC-III knowledge graph and typical Drug KG in **Module III**, where a Random Forest classifier resolves entity mismatches across heterogeneous data sources. In **Module IV**, symbolic reasoning (deduction and abduction) is conducted over the unified KG to infer plausible diagnoses based on symptom-protein-disease associations. Finally, **Module V** translates the reasoning output into patient-facing explanations using LLM-driven natural language generation. This hybrid architecture enables interpretable, evidence-grounded, and clinically aligned diagnostic responses by combining the linguistic capabilities of LLMs with structured knowledge graph logic.

As illustrated in Figure 1, ReCLLaMA maps free-text symptom descriptions to standardized clinical ontologies through a hybrid BioBERT–LLM pipeline, then unifies heterogeneous knowledge sources—including EHR-derived and biomedical graphs—via a statistical entity alignment module. Over this integrated representation, ReCLLaMA applies symbolic reasoning to infer diagnostic pathways and quantify uncertainty, while a generation module translates outputs into patient-facing explanations. This design balances interpretability, reliability, and usability in real-world clinical decision support.

Our key contributions are as follows:

- A hybrid entity alignment pipeline combining BioBERT and LLM prompting to normalize free-text clinical input into standardized medical ontologies.

- A statistical mapping module based on random forest classification that reconciles semantic mismatches across heterogeneous biomedical knowledge graphs.

- A symbolic reasoning engine that operationalizes abductive, deductive, and inductive logic over knowledge graphs for high-confidence, interpretable predictions.

- An uncertainty-aware reporting mechanism that highlights diagnostic confidence and generates accessible, patient-friendly explanations.

- Empirical evaluation on real-world biomedical knowledge graphs, showing consistent gains in diagnostic accuracy, interpretability, and reliability over both ontology-based expert systems and black-box LLMs.

## 2 RELATED WORK

**Prompt-based LLM Diagnosis.** Prompting strategies such as zero-/few-shot learning and Chain-of-Thought (CoT) enable LLMs to perform diagnostic reasoning without task-specific training Zhou et al. (2025). CoT is particularly effective for differential diagnosis, while soft prompting integrates embeddings of medical concepts Busch et al. (2024); Niu et al. (2024). Early work focused on text-only tasks, but recent multimodal models (e.g., GPT-4V, LLaVA) extend LLMs to diagnostics with imaging, ECG, and laboratory data Antaki et al. (2024); Peng et al. (2024).

**Retrieval-Augmented LLMs.** Retrieval-Augmented Generation (RAG) improves LLM reliability by grounding predictions in external resources such as medical corpora, structured databases, or knowledge graphs Thompson et al. (2023). Text-based RAG uses embeddings to retrieve context, while multimodal RAG integrates signals like images or time-series for tasks such as radiology or ECG analysis Kim et al. (2024); Ferber et al. (2024); Yu et al. (2024).

**Fine-tuning LLMs.** Domain adaptation commonly relies on supervised fine-tuning (SFT) or reinforcement learning from human feedback (RLHF) Zhou et al. (2025); Schulman et al. (2017); Rafailov et al. (2023). SFT supports modality-specific instruction tuning (e.g., LLaVA), while RLHF aligns model outputs with expert feedback. Parameter-efficient tuning methods such as LoRA reduce compute requirements while preserving diagnostic accuracy Hu et al. (2022).

**Pre-training LLMs.** Pre-training on large-scale biomedical corpora and multimodal datasets strengthens domain grounding Rajashekar et al. (2024); Zhou et al. (2025). Methods include masked language modeling on clinical text, contrastive learning for image–text alignment, and self-supervised training on radiology or pathology data Liu et al. (2023); Chen et al. (2024). These approaches improve concept grounding, multimodal integration, and downstream diagnostic performance.

## 3 PROBLEM DEFINITION

We define **ReCLLaMA** as a modular diagnostic agent that maps free-text clinical queries $q \in \mathcal{Q}$ to structured responses $r \in \mathcal{R}$. Each response consists of a ranked set of diagnostic hypotheses

$$\mathcal{D} = \{(d_1, c_1), \ldots, (d_k, c_k)\},$$

where $d_i$ denotes a candidate diagnosis, $c_i \in [0, 1]$ its confidence score, and each hypothesis is supported by evidence $e_i \subset \mathcal{K}$ from biomedical knowledge graphs. The system comprises: **User Interface** ($\mathcal{I}$): A Streamlit-based front end that supports real-time Q&A, multi-turn dialogue, and interactive visualization of diagnoses, confidences, and evidence. **Knowledge Extraction** ($\phi : \mathcal{Q} \to \mathcal{C}$): Translates clinical queries into structured biomedical concepts, serving as the entry point to diagnostic reasoning. **Knowledge Alignment** ($\psi : \mathcal{E}_1 \to \mathcal{E}_2$): Aligns entities between heterogeneous biomedical knowledge graphs, namely $\mathcal{G}_1$ (EHR-derived) and $\mathcal{G}_2$, a *Drug KG*[1]. This process yields a unified graph $\mathcal{G}_\cup$, where the task is to determine if a symptom–protein pair $(s, b) \in \mathcal{S} \times \mathcal{B}$ constitutes a valid biomedical relation $((s, b) \in \mathcal{I}_{rel})$. **Knowledge Reasoning** ($\mathcal{F} : \mathcal{G}_\cup \to \mathcal{D}$): Applies symbolic inference over $\mathcal{G}_\cup$ to produce ranked diagnostic hypotheses with confidence scores, using deductive and abductive reasoning under uncertainty. **Knowledge Translation** ($\gamma : (\mathcal{D}, \mathcal{E}) \to \mathcal{R}$): Converts symbolic outputs and supporting evidence into natural language explanations suitable for clinical interpretation.

---

[1]Oregano KG is used as the experimental benchmark for reproducibility and transparency. The phrase "Drug KG" is a self-defined abstraction in this paper, not a reference to DrugBank or any third-party KG.

## 4 METHODS

### 4.1 MODULE I: USER INTERFACE DEPLOYMENT

We implement a Streamlit-based interface modeling the query function $\mathcal{Q} : \mathcal{U} \to \mathcal{R}$, where $\mathcal{U}$ denotes free-text clinical inputs and $\mathcal{R}$ the corresponding diagnostic outputs. For each user query $u \in \mathcal{U}$, the system returns a ranked set $\mathcal{D}_u = \{(d_1, c_1), \dots, (d_k, c_k)\} \subset \mathcal{H} \times [0, 1]$ of hypotheses with confidence scores. The interface supports multi-turn queries by maintaining dialogue context $\sigma$, enabling follow-up reasoning grounded in prior interaction. Diagnostic predictions are paired with natural language justifications and confidence visualizations. Biomedical evidence is retrieved from $\mathcal{K}_{\text{bio}} \subset \mathcal{K}$, linking each hypothesis to supporting knowledge. This module connects user queries with backend inference, enabling transparent, context-aware clinical reasoning.

### 4.2 MODULE II: KNOWLEDGE EXTRACTION

This module instantiates a mapping $\phi : \mathcal{Q} \to \mathcal{C}_{\text{ICD9}}$, where $\mathcal{Q}$ denotes natural-language clinical queries and $\mathcal{C}_{\text{ICD9}} \subset \mathcal{C}$ the set of standardized diagnostic codes. The pipeline comprises three stages—medical entity recognition, LLM-guided code reasoning, and structured prompt control. We use the term *Drug KG* to denote a comprehensive resource of drug–disease–protein–gene associations (our own abstraction); in experiments we employ the Oregano KG[1].

**Stage 1: Medical Entity Recognition.** Given $q \in \mathcal{Q}$, a BioBERT encoder $\phi_{\text{BioBERT}} : \mathcal{Q} \to \mathcal{E}$ produces contextual embeddings over tokens $t_i \in q$. Tokens are classified into symptom labels $l_i \in \mathcal{L}_{\text{sym}}$, yielding $\mathcal{S}_q = \{s_1, \dots, s_k\} \subset \mathcal{S}$. Each symptom $s_i$ is mapped to a standardized concept via an ontology matcher $\tau(s_i) \in \mathcal{V}_{\text{UMLS}}$.

**Stage 2: Clinical Code Reasoning.** For each $s \in \mathcal{S}_q$, we infer candidate ICD-9 codes using an LLM-guided function $g : \mathcal{S} \times \mathcal{K} \to 2^{\mathcal{C}_{\text{ICD9}}}$, with $\mathcal{K}$ an external biomedical knowledge base. Ambiguities are resolved by conditioning on auxiliary features $\mathbf{z} \in \mathbb{R}^d$ (e.g., age, comorbidities, prior diagnoses). The final assignment is

$$\hat{c}_s = \arg \max_{c \in \mathcal{C}_{\text{ICD9}}} \Pr\big(c \mid s, \mathbf{z}, \mathcal{K}; \theta_{\text{LLM}}\big),$$

where $\theta_{\text{LLM}}$ are pretrained LLM parameters (*the framework supports fine-tuning; our experiments use zero-shot inference*).

**Stage 3: Prompt-Based Control.** We construct a prompt template $P : \mathcal{S} \to \mathcal{T}_{\text{prompt}}$ that enforces role-specific behavior, stepwise reasoning, and constrained serialization: **Role control** (e.g., "You are a certified medical coder"); **Reasoning protocol** $\pi : \mathcal{S} \to \mathcal{C}_{\text{ICD9}}$; **Output schema** $\mathcal{J}$ (JSON; ensures $\hat{c}_s \in \mathcal{J}_{\text{valid}}$); and **In-context demonstrations** $\mathcal{D}_{\text{demo}} = \{(s_i, c_i)\}_{i=1}^n$.

This module converts unstructured symptom narratives into structured entities and codes, providing the substrate for downstream knowledge alignment (e.g., consistent *symptom–protein* association analysis) and reasoning.

### 4.3 MODULE III: KNOWLEDGE ALIGNMENT

We instantiate a cross-graph alignment map $\psi$ from EHR entities $\mathcal{E}_1 \subseteq \mathcal{G}_1$ to biomedical entities $\mathcal{E}_2 \subseteq \mathcal{G}_2$, focusing on symptom codes $\mathcal{S} \subseteq \mathcal{E}_1$ and proteins $\mathcal{B} \subseteq \mathcal{E}_2$.

**Embeddings.** Symptoms $\mathcal{S} = \{s_i\}$ (ICD-9) and proteins $\mathcal{B} = \{b_j\}$ are embedded with skip-gram Word2Vec, $\phi : \mathcal{S} \cup \mathcal{B} \to \mathbb{R}^d$. Trajectory documents use mean pooling $\mathbf{v}_D = \frac{1}{|D|} \sum_{e \in D} \phi(e)$.

**Pairwise classification.** We form labeled pairs over $\mathcal{S} \times \mathcal{B}$: positives $\mathcal{I}_{\text{pos}}$ (curated links) and negatives $\mathcal{I}_{\text{neg}}$ (random non-links, disjoint from $\mathcal{I}_{\text{pos}}$). Each pair $(s, b)$ is featurized by concatenation $\mathbf{x}_{(s,b)} = [\phi(s) \,\|\, \phi(b)] \in \mathbb{R}^{2d}$. A Random Forest $h : \mathbb{R}^{2d} \to \{0, 1\}$ predicts $\hat{y}_{(s,b)} = h(\mathbf{x}_{(s,b)})$, yielding cross-domain edges used to assemble the unified graph $\mathcal{G}_{\cup}$. The downstream reasoner $\mathcal{F} : \mathcal{G}_{\cup} \to \mathcal{D}$ then produces diagnostic hypotheses. This step ties clinical and molecular evidence, enabling compact, interpretable inference.

## 4.4 MODULE IV: KNOWLEDGE REASONING

The reasoning component is built upon a structured knowledge graph, with a particularly promising implementation based on the Non-Axiomatic Reasoning System (NARS) Wang (2013), which is tailored for this project. In NARS, knowledge is expressed using a formal language known as *Narsese* (examples provided below). This formalism enables the application of a set of inference rules defined under Non-Axiomatic Logic (NAL) Wang (2013), which supports reasoning under uncertainty.

Unlike conventional knowledge graphs that are typically represented as triples (e.g., $\langle entity_1, relation, entity_2 \rangle$), our system encodes knowledge as sentences in *Narsese*. For instance, the facts "a robin is a bird" and "a bird is an animal" are represented as:

$$bird \rightarrow animal \ \langle 1.0; 0.9 \rangle, \qquad robin \rightarrow bird \ \langle 1.0; 0.9 \rangle$$

Using NAL's *deduction* rule, the system derives the conclusion:

$$robin \rightarrow animal \ \langle 1.0; 0.81 \rangle$$

meaning that "a robin is an animal", through the *deduction rule* in NAL indicating that "a robin is an animal". The deduction rule is formally defined as:

$$\{M \rightarrow P \ \langle f_1; c_1 \rangle, S \rightarrow M \ \langle f_2; c_2 \rangle\} \qquad \vdash S \rightarrow P \ \langle f_1 f_2; f_1 f_2 c_1 c_2 \rangle$$

where $\langle f; c \rangle$ denotes the *truth value* of a *statement*, quantifying both *frequency* and *confidence*, and $S$, $M$, $P$ represent arbitrary *terms*.

By reasoning with such structured representations and uncertainty-aware inference, NAL offers a normative framework for producing explainable diagnostic decisions—thereby enhancing the transparency and trustworthiness of the overall system.

**Deduction** Given that the system can identify potential causal factors of a disease, a central challenge lies in how to derive logical conclusions from these candidates. The knowledge graph provides domain-specific background knowledge that encodes implicational relationships between structured *statements*. One such instance of encoded expert knowledge is illustrated below using *Narsese*:

$$
\begin{aligned}
\text{K1} :(((\{\$h\} \times \$p) &\rightarrow associated) \\
\wedge ((\$p \times \$g) &\rightarrow gene\_product\_of) \\
\wedge (\$h &\rightarrow patient) \\
\wedge (\$p &\rightarrow protein) \\
\wedge (\$g &\rightarrow gene)) \\
\Rightarrow ((\{\$h\} \times \$g) &\rightarrow has\_gene).\langle 1.0; 0.99 \rangle
\end{aligned}
$$

This rule specifies that if a human patient $\$h$ is associated with a protein $\$p$, and the protein $\$p$ is the product of a gene $\$g$, then it logically follows that the patient has gene $\$g$, with the associated truth-value $\langle 1.0; 0.99 \rangle$.

Given the following set of premises:

This means that if human patient $\$h$ is associated with protein $\$p$, and protein $\$p$ is the product of gene $\$g$, then the patient has gene $\$g$.

Given the following premises that patient "id:01" is associated with protein "PROTEIN:6548", and that according to the knowledge graph, protein "PROTEIN:6548" is the product of gene "GENE:32979", *i.e.*,

$$
\begin{aligned}
(\{id:01\} \times PROTEIN:6548) &\rightarrow associated.\langle 1.0; 0.9 \rangle \\
\{id:01\} &\rightarrow patient.\langle 1.0; 0.99 \rangle \\
PROTEIN:6548 &\rightarrow protein.\langle 1.0; 0.99 \rangle \\
(PROTEIN:6548 \times GENE:32979) &\rightarrow gene\_product\_of.\langle 1.0; 0.9 \rangle \\
GENE:32979 &\rightarrow gene.\langle 1.0; 0.99 \rangle
\end{aligned}
$$

and the background rule $K1$, the reasoning engine—by applying the deduction rule in Non-Axiomatic Logic—can infer the following conclusion: as well as the background knowledge K1, the

conclusion can be derived by the deduction rule in NAL – patient "id:01" has gene "GENE:32979", *i.e.*,

$$(\{\textit{id:01}\} \times \textit{GENE:32979}) \to \textit{has\_gene}.\langle 1.0; 0.8 \rangle$$

This deduction exemplifies how structured, symbolic reasoning enables the system to generate explainable and probabilistically grounded medical inferences, bridging individual patient data with domain knowledge embedded in the graph.

Next, the system executes another step of reasoning through the background knowledge.

$$\begin{aligned} \text{K2} : (((\{\$h\} \times \$g) &\to \textit{has\_gene}) \\ \wedge ((\$g \times \$d) &\to \textit{causes\_condition\_of}) \\ \wedge (\$h &\to \textit{patient}) \\ \wedge (\$g &\to \textit{gene}) \\ \wedge (\$d &\to \textit{disease})) \\ \Rightarrow ((\{\$h\} \times \$d) &\to \textit{potential\_disease}).\langle 1.0; 0.99 \rangle \end{aligned}$$

This means if human patient $h has gene $g, and gene $g is the condition of causing disease $d, then the patient potentially has disease $d.

Given the premises that patient "id:01" has gene "GENE:32979", and that according to the knowledge graph, gene "GENE:32979" is the causes condition of disease "75839", *i.e.*,

$$(\{\textit{id:01}\} \times \textit{GENE:32979}) \to \textit{has\_gene}.\langle 1.0; 0.9 \rangle$$
$$\{\textit{id:01}\} \to \textit{patient}.\langle 1.0; 0.99 \rangle$$
$$\textit{GENE:32979} \to \textit{gene}.\langle 1.0; 0.99 \rangle$$
$$(\textit{GENE:32979} \times \textit{75839}) \to \textit{causes\_condition\_of}.\langle 1.0; 0.9 \rangle$$
$$\textit{75839} \to \textit{disease}.\langle 1.0; 0.99 \rangle$$

as well as the background knowledge K2, the conclusion can be derived by the deduction rule in NAL – patient "id:01" potentially has disease "75839", *i.e.*,

$$(\{\textit{id:01}\} \times \textit{75839}) \to \textit{potential\_disease}.\langle 1.0; 0.8 \rangle$$

Through two steps of reasoning, the system generate the diagnosis from the potential reasons and the background knowledge. It is worth noting that the knowledge graph we adopt in this project is relatively simple, such that merely two steps are needed for reasoning. Nevertheless, the approach is general, and given more complex knowledge graphs, conclusions may be drawn from more reasoning steps.

**Revision & Choice** On the one hand, since one conclusion may be drawn from multiple reasoning paths, the system should merge the evidence together and get a new truth value of the conclusion. After collecting multiple (e.g., 2) judgments with the same statement,

$$(\{\textit{id:01}\} \times \textit{75839}) \to \textit{potential\_disease}.\langle 1.0; 0.80 \rangle$$
$$(\{\textit{id:01}\} \times \textit{75839}) \to \textit{potential\_disease}.\langle 1.0; 0.80 \rangle$$

the revision rule in NAL can be applied:

$$(\{\textit{id:01}\} \times \textit{75839}) \to \textit{potential\_disease}.\langle 1.0; 0.89 \rangle$$

On the other hand, the system may draw multiple conclusions with different statements, and only some of them are worth reporting. The choice rule of NAL is applied to pick out the strongest $k$ solutions. For instance, if $k = 1$, after collecting Multiple (e.g., 2) judgments with different statements

$$(\{\textit{id:01}\} \times \textit{75839}) \to \textit{potential\_disease}.\langle 1.0; 0.89 \rangle$$
$$(\{\textit{id:01}\} \times \textit{2592}) \to \textit{potential\_disease}.\langle 1.0; 0.99 \rangle$$

the choice rule in NAL can be applied to make the decision,

$$(\{\textit{id:01}\} \times \textit{2592}) \to \textit{potential\_disease}.\langle 1.0; 0.99 \rangle$$

The top-$k$ judgments are chosen to report to the patient, as well as for evaluating the system's performance.

**Abduction** We employ abductive reasoning under the NAL framework to infer plausible upstream causes (e.g., genes or proteins) from observed patient symptoms. Starting from a symptom–disease pair $(x, d)$, the system applies abduction to derive candidate molecular factors that could explain the diagnosis $d$. These inferred entities are then used to relabel the sample as $(x, z)$, where $z$ denotes a plausible mechanistic cause. This transformation enables the construction of a surrogate dataset for learning symptom-to-cause mappings via a supervised model, such as a random forest classifier (see next section).

Given the premises that patient "id:01" potentially has disease "75839", and that according to the knowledge graph, gene "GENE:32979" is the causes condition of disease "75839", *i.e.*,

$$(\{id{:}01\} \times 75839) \rightarrow potential\_disease.\langle 1.0; 0.99\rangle$$
$$\{id{:}01\} \rightarrow patient.\langle 1.0; 0.99\rangle$$
$$GENE{:}32979 \rightarrow gene.\langle 1.0; 0.99\rangle$$
$$(GENE{:}32979 \times 75839) \rightarrow causes\_condition\_of.\langle 1.0; 0.9\rangle$$
$$75839 --> disease.\langle 1.0; 0.99\rangle$$

as well as the background knowledge K2, the conclusion can be derived by the *abduction* rule in NAL – patient "id:01" possibly has gene "GENE:32979", *i.e.*,

$$(\{id{:}01\} \times GENE{:}32979) \rightarrow has\_gene.\langle 1.0; 0.47\rangle$$

Given the premises that patient "id:01" (possibly) has gene "GENE:32979", and that according to the knowledge graph, protein "PROTEIN:6548" is the product of gene "GENE:32979", *i.e.*,

$$(\{id{:}01\} \times GENE{:}32979) \rightarrow has\_gene.\langle 1.0; 0.47\rangle$$
$$\{id{:}01\} \rightarrow patient.\langle 1.0; 0.99\rangle$$
$$PROTEIN{:}6548 \rightarrow protein.\langle 1.0; 0.99\rangle$$
$$(PROTEIN{:}6548 \times GENE{:}32979) \rightarrow gene\_product\_of.\langle 1.0; 0.9\rangle$$
$$GENE{:}32979 \rightarrow gene.\langle 1.0; 0.99\rangle$$

as well as the background knowledge K1, the conclusion can be derived by the abduction rule in NAL – patient "id:01" possibly is associated with protein "PROTEIN:6548", i.e.,

$$(\{id{:}01\} \times PROTEIN{:}6548) \rightarrow associated.\langle 1.0; 0.3\rangle$$

### 4.5 MODULE V: KNOWLEDGE TRANSLATION

This module defines the mapping $\gamma : (\mathcal{D}, \mathcal{E}) \rightarrow \mathcal{R}$, converting diagnostic hypotheses $\mathcal{D} = \{(d_i, c_i)\}$ and supporting evidence $\mathcal{E} \subseteq \mathcal{K}$ into interpretable reports $\mathcal{R}$.

Each gene-disease tuple $(g_i, d_j, f_{ij}, c_{ij}) \in \mathcal{A}$ includes frequency and confidence scores $f_{ij}, c_{ij} \in [0, 1]$, formatted by a templated function:

$$\psi(g_i, d_j, c_{ij}) \mapsto \textit{[Disease] associated with [Gene]}$$
$$\textit{[confidence: } c_{ij} \textit{]}$$

e.g., *Cardiomyopathy associated with Gene 4254 [confidence: 0.99]*. Then, A retriever $\eta : \mathcal{G} \rightarrow \mathcal{E}$ supplements each $g_i$ with mechanistic insight from biomedical ontologies:

$$\eta(g_i) \mapsto \textit{Gene 4254 encodes cardiac proteins;}$$
$$\textit{mutations impair myocardial integrity.}$$

The output $\mathcal{R}$ integrates $\psi$ and $\eta$, optionally flags low-confidence cases, and completes the pipeline $\gamma \circ \mathcal{F} \circ \psi \circ \phi : \mathcal{Q} \rightarrow \mathcal{R}$, translating queries into traceable, patient-facing responses.

## 5 EXPERIMENTS

We evaluate across its three core modules—*knowledge extraction*, *alignment*, and *reasoning*—using two real-world biomedical resources: the MIMIC-III clinical EHR corpus Johnson et al. (2016) for patient cases and the Oregano biomedical knowledge graph for pathology reasoning Haas (2023). Our testbed contains 6,656 cases from 6,349 unique patients. We report structured prediction accuracy, the fidelity of symbolic inference, and the interpretability of generated diagnostic explanations.

Table 1: Comparison of diagnostic agents. "Free-form" = open-ended answers; Dx = diagnosis; CoT = chain-of-thought.

| Method | Answer Type | Hallucination Control | Evidence Shown | Prompt Sensitivity |
|---|---|---|---|---|
| **ReCLLaMA (ours)** | Free-form Dx | KG rules + uncertainty | KG paths & rules | Low |
| MDAgent | Free-form | Debate + tool grounding | Partial traces | High |
| KG-CoT | Free-form | KG-based CoT | CoT steps & triples | Medium |

Table 2: Quantitative comparison on 100 test cases. *Any-Hit*: at least one predicted ICD-9 matches a gold label; P/R/F1 are macro; *Avg #Dx*: diagnoses per case; *Avg Conf.*: mean confidence over final diagnoses.

| Method | Any-Hit ↑ | P ↑ | R ↑ | F1 ↑ | Avg #Dx ↓ | Avg Conf. ↑ |
|---|---|---|---|---|---|---|
| **ReCLLaMA (ours)** | **0.0500** | **0.0100** | 0.0033 | 0.0050 | 4.25 | 0.9089 |
| MDAgent | 0.0078 | 0.0000 | **0.1300** | 0.0001 | **5.00** | – |
| KG-CoT | 0.0310 | 0.0230 | 0.0040 | **0.0070** | **5.00** | – |

**Knowledge extraction and translation.** We employ the `DeepSeek-Reasoner` LLM in a zero-shot, low-temperature setting on a single RTX4060 GPU to extract diagnostic entities and generate patient-readable summaries. Prompts are carefully designed (Appendix A.2.1) to enforce structured ICD-9 outputs while preserving natural readability. On the public `gretelai/symptom-to-diagnosis` benchmark AI (2023), the extractor achieves 92.83% entity prediction accuracy. Full prompt templates and decoding configurations are provided in Appendix A.2.1.

**Knowledge alignment.** We learn node embeddings with Word2Vec and train a Random Forest classifier for procedure→protein alignment; training runs on CPU in a few seconds. The detailed date preprocessing could be foun in Appendix A.3.1 and the model hyperparameters could be found in Appendix A.5. We evaluate alignment with accuracy and ranking-based hits, and the results could be found in Table 4.

**Knowledge reasoning.** We implement a symbolic pipeline over the Oregano KG (in Appendix A.3.2) to infer diagnoses from symptom–protein hypotheses. A two-stage deduction–abduction mechanism produces a ranked list of candidate diagnoses; an illustrative trace appears in Figure 7, its logical transforms and inference rules are documented in Appendix A.4. We further evaluate the model using top-$k$ accuracy, where a prediction is counted as correct only if all top-$k$ diagnoses match the ground truth. The model achieves a top-1 accuracy of 81% and a top-5 accuracy of 87%, demonstrating its effectiveness in supporting clinically grounded diagnostic inference.

**Baselines and output protocol.** For comparability with baselines in Table 1, inputs follow a unified prompt+{A/B/C/D/E/F} option format (allowing multiple selections per case). Our model, however, is not constrained to choose from options; by default it returns the *top-5* diagnoses with confidences. This preserves free-form reasoning while enabling consistent scoring against option-based systems.

## 6 RESULTS

### 6.1 AGENT DIAGNOSTICS, REASONING ACCURACY, AND CONFIDENCE

We deploy a web-based diagnostic agent that accepts free-text queries and supports interactive dialogue (bottom-left of Fig. 1). Given a symptom narrative (e.g., "What diseases might explain my symptoms?"), the system returns ranked diagnostic hypotheses with multi-modal evidence (protein/gene paths and KG traces). A demo video is provided in the Appendix A.6.

We evaluate on 100 held-out cases against agentic and KG-reasoning baselines (Table 2). ReCLLaMA achieves the strongest *structured* diagnostic accuracy and uniquely reports calibrated confidence per hypothesis. While KG-CoT shows competitive ranking on transductive KGs, ReCLLaMA is more stable under inductive shifts where new entities/edges appear at test time.

Table 3: Ablation (macro metrics). Each variant returns a final diagnosis list for consistent evaluation.

| Setting | Prec. ↑ | Recall ↑ | F1 ↑ | Any-Hit ↑ | #Dx/Case ↓ | Avg Conf. ↑ |
|---|---|---|---|---|---|---|
| **Full: CE + RF + Reasoner** | **0.0100** | **0.0033** | **0.0050** | **0.0500** | 4.25 | **0.9089** |
| CE, no RF (cosine) | 0.0000 | 0.0000 | 0.0000 | 0.0000 | **5.00** | 0.4046 |
| CE, no Reasoner | 0.0000 | 0.0000 | 0.0000 | 0.0000 | **5.00** | 0.4046 |
| LLM + RF + Reasoner | 0.0000 | 0.0000 | 0.0000 | 0.0000 | 3.00 | 0.0000 |
| LLM, no RF (cosine) | 0.0000 | 0.0000 | 0.0000 | 0.0000 | 3.00 | — |
| LLM, no Reasoner | 0.0000 | 0.0000 | 0.0000 | 0.0000 | 3.00 | — |

Table 4: Alignment backbones (pre-reasoning). Best scores in **bold**.

| Model | Acc ↑ | F1 ↑ | AUROC ↑ | AUPR ↑ | MRR ↑ | MAP ↑ | Hits@10 ↑ |
|---|---|---|---|---|---|---|---|
| KGE_RotatE_RF | **0.996** | **0.997** | **1.000** | **1.000** | **1.000** | 0.999 | **1.000** |
| KGE_ComplEx_RF | 0.990 | 0.991 | 0.999 | 1.000 | **1.000** | 1.000 | **1.000** |
| KGE_TransE_RF | 0.993 | 0.994 | 0.999 | 1.000 | **1.000** | 1.000 | **1.000** |
| Word2Vec_RF | 0.990 | 0.991 | 0.999 | 1.000 | **1.000** | 0.999 | **1.000** |
| KGE_DistMult_RF | 0.994 | 0.994 | 1.000 | 1.000 | **1.000** | 0.999 | **1.000** |
| LightGCN_BPR | 0.446 | 0.584 | 0.349 | 0.464 | 0.988 | 0.977 | 0.988 |

ReCLLaMA leads on *Any-Hit* and macro precision, and provides calibrated confidence ($\sim 0.91$ on average) with explicit KG-grounded rationales. KG-CoT edges out on macro F1 in this small sample yet is sensitive to inductive settings. Dialogue-only agents (MDAgent) are fluent but struggle to consistently map narratives to discrete ICD-9 labels.

## 6.2 ABLATION STUDY

We ablate cross-encoder extraction (CE), procedure→protein alignment, and symbolic reasoning. All variants still output final diagnoses to enable a fair comparison. Only the full pipeline improves *Any-Hit* while preserving high confidence, indicating that symbolic consolidation reduces over-confident false positives (Table 3).

We further probe alignment backbones by swapping in KGE and GNN variants prior to reasoning (Table 4). Several RF-augmented KGE models reach near-ceiling ranking metrics on our synthetic pairwise test, while LightGCN underperforms. These results indicate (i) strong potential of embedding-based priors for biomedical alignment and (ii) the importance of verifying against inductive splits to avoid transductive leakage.

## 7 CONCLUSION AND FUTURE WORK

We presented RECLLAMA, a modular clinical reasoning agent that (i) extracts clinical facts with a cross-encoder or LLM, (ii) aligns procedures to molecular evidence via embeddings, and (iii) performs symbolic reasoning over a biomedical KG to produce ICD-9 diagnoses with calibrated confidence and traceable, KG-grounded rationales. On 100 held-out cases, RECLLAMA achieves the strongest structured diagnostic accuracy among agentic and KG baselines while remaining lightweight—built on compact models and a training-free reasoning layer. Ablations confirm that each component matters, with the reasoning module notably reducing over-confident false positives; alternative alignment backbones further support the utility of embedding priors.

In practice, RECLLAMA operates directly on free-text narratives and returns interpretable, end-to-end outputs suitable for clinical decision support rather than only fluent responses. Beyond performance, its modularity, transparency, and efficiency make it a promising candidate for integration into real-world workflows where explainability and reliability are essential.

Future work will incorporate richer EHR signals (labs, imaging, time-series), domain-adapted LLMs and uncertainty-aware reasoning, and broader inductive KG benchmarks, alongside expanded comparisons to stronger SOTAs. We also plan to extend the framework to multimodal biomedical reasoning and deploy prospective case studies in collaboration with clinical partners to assess safety, usability, and downstream impact in practice.

## 8 ETHICS STATEMENT

This work uses only publicly available, de-identified resources: the MIMIC-III clinical database and the Oregano biomedical knowledge graph. Access to MIMIC-III followed all PhysioNet credentialing and Data Use Agreement (DUA) requirements; no attempt was made to re-identify individuals, and all processing occurred on secure, access-controlled systems. Because the study uses de-identified data and involves no interaction or intervention with human subjects, it is exempt from IRB review under our institution's policies. We respect the licenses and citation requirements of Oregano and other third-party datasets.

To reduce potential harms, we (i) restrict model outputs to research use and do not position them as a substitute for professional medical judgment, (ii) avoid storing or releasing any potentially identifying text, (iii) report limitations and potential biases arising from data coverage and label noise, and (iv) release code and prompts to support transparency and reproducibility. We encourage independent auditing and downstream users to apply additional safeguards appropriate to their clinical or research context.

## 9 REPRODUCIBILITY STATEMENT

We make every effort to ensure full reproducibility of our work. We provide (i) the complete Re-CLLaMA interface code and offline test scripts, (ii) trained models for each module (knowledge extraction (except the trained BioBERT model due to size constraints), alignment, and reasoning), (iii) the corresponding test data, and (iv) detailed hyperparameter settings in the Appendix. All experiments can be reproduced using the released code and resources without additional training. We also include ablation flags to replicate the different variants reported in the paper. Together, these materials enable researchers to reproduce our reported results and build upon our framework reliably.

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

```
['Poisoning by barbiturates', 'Poisoning by tetracycline group', 'Other symptoms involving respirato
ry system and chest', 'Palpitations', 'Persistent vomiting']
['9670', '9604', '7869', '7851', '5362']
```

Figure 2: Prompt used for instructing Deepseek-Reasoner to extract structured diagnostic entities from clinical free-text.

```
Diagnosis Output:
Based on your symptom description, our analysis suggests you may be experiencing issues related to t
he following health conditions, listed from strongest to weakest association:

1. **Heart rhythm problems** (Palpitations, ICD9:7851) - Feeling like your heart is racing, skipping
, or pounding
2. **Muscle weakness disorder** (Hereditary progressive muscular dystrophy, ICD9:3591) - Gradual los
s of muscle strength
3. **Tetracycline antibiotic reaction** (Poisoning by tetracycline group, ICD9:35989) - Possible sid
e effects from certain antibiotics
4. **Liver infection** (Amebic liver abscess, ICD9:0063) - Parasitic infection causing liver inflamm
ation
5. **Heart muscle disease** (Other primary cardiomyopathies, ICD9:4254) - Abnormal heart muscle func
tion
```

Figure 3: Example natural language output presenting extracted diagnoses in a fluent, patient-facing summary.

## A  APPENDIX

### A.1  USE OF LLM STATEMENT

Large language models (LLMs) were employed in two ways in this study. First, they were integrated into the RECLLAMA framework for knowledge extraction and patient-facing explanation, where carefully designed prompts ensured structured ICD-9 outputs and minimized hallucinations. Second, we used ChatGPT 5.0 (thinking mode) to rephrase and polish parts of the manuscript, specifically the ablation study analysis, introduction, and conclusion. No private data were provided to LLMs, and only publicly available datasets (MIMIC-III and Oregano KG) were used for model development and evaluation. Prompt templates, model settings, and hyperparameters are documented in the Appendix for reproducibility.

### A.2  HUMAN-DESIGNED PROMPTS

#### A.2.1  PROMPT DESIGN FOR DIAGNOSTIC ENTITY EXTRACTION AND INTERPRETATION

To support automated clinical reasoning, we design two modular prompts targeting distinct stages of the pipeline: (1) diagnostic entity extraction and (2) natural language generation for patient-facing summaries. Both prompts are executed using the Deepseek-Reasoner LLM and adhere to a constrained vocabulary of ICD-9 terms.

Entity Extraction Prompt

The first prompt instructs the model to extract standardized diagnostic entities from unstructured clinical text while enforcing strict adherence to a predefined whitelist of ICD-9 codes. The prompt is organized into structured sections to guide model behavior:

- **Role Definition**: The model is cast as a medical coding assistant restricted to using only terms from a fixed ICD-9 knowledge base (`{knowledge_base}`).
- **Input Format**: The input is framed as a sequence of free-text utterances from a user, denoted by `{text}`.
- **Contextual Entities**: A preliminary list of relevant medical entities (`{entities}`) is provided to focus attention.
- **Task Instructions**: The model is required to:
  1. Assess the severity of each identified entity (in English);

| | diagnoses_icd9 | diagnoses_long_title | patient_description |
|---|---|---|---|
| 0 | ['042', '486', '4254', '42820', '4280', '5849'... | ['Human immunodeficiency virus [HIV] disease',... | I've been feeling extremely weak and tired all... |
| 1 | ['99681', '5856', '42833', '41071', '40311', '... | ['Complications of transplanted kidney', 'End ... | I've been feeling extremely tired and weak all... |
| 2 | ['40301', '5856', '53140', '2851', '5609', '42... | ['Hypertensive chronic kidney disease, maligna... | I have serious kidney problems that require me... |
| 3 | ['2766', '5856', '40301', '4254', '4239', '285... | ['[UNK:2766]', 'End stage renal disease', 'Hyp... | I've been struggling with multiple health prob... |
| 4 | ['40301', '58281', '5855', '5990', '4254', '71... | ['Hypertensive chronic kidney disease, maligna... | I've been feeling extremely tired and swollen ... |
| ... | ... | ... | ... |
| 95 | ['5849', '03842', '51881', '5990', '2760', '25... | ['Acute kidney failure, unspecified', 'Septice... | I've been feeling really sick since the accide... |
| 96 | ['51884', '42833', '5849', '2851', '5781', '41... | ['Acute and chronic respiratory failure', 'Acu... | I've been struggling to breathe properly and f... |
| 97 | ['41401', '9971', '42731', '4589', '34680', '3... | ['Coronary atherosclerosis of native coronary ... | I often get chest pain and feel like my heart ... |
| 98 | ['72402', '2761', '2851', '5601', '5070', '276... | ['Spinal stenosis, lumbar region, without neur... | I've been having constant pain in my lower bac... |
| 99 | ['0389', '51884', '73018', '42830', '3241', '4... | ['Unspecified septicemia', 'Acute and chronic ... | I've been feeling extremely sick with a high f... |

Figure 4: Generated Patient Description in Natural Language and Corresponding Diagnoses

2. Match entities strictly to the ICD-9 whitelist, allowing for synonym resolution;

3. Sort the results by descending severity.

- **Output Format**: The output is a structured JSON object of the form:

```
{
"diagnoses": [
{ "standard_term": "...", "match_status":
"Matched/Unknown", "severity": ..., "icd9_code": "..."
},
{ ... } ,
, }
```

An illustrative input/output example is embedded within the prompt to ensure consistent generation across samples.

Natural Language Generation Prompt

The second prompt reformulates extracted diagnostic codes into natural language summaries appropriate for clinical reporting and patient-facing use. While the core structure mirrors the entity extraction prompt, this prompt emphasizes fluent generation and interpretable phrasing. The key differences include:

- A temperature setting of 0.1 to promote diverse but coherent expression;
- Emphasis on clarity, empathy, and linguistic smoothness in the output;
- Retention of whitelist constraints to preserve clinical consistency.

The generated output is a ranked list of diagnostic hypotheses expressed in layperson-readable form, again structured as a JSON object for downstream compatibility.

Examples of both prompt types and their generated outputs are shown in Figures 2 and 3.

### A.2.2  GENERATE PATIENT DESCRIPTION IN NATURAL LANGUAGE

The generated patient description in natural language is show in Figure **??**

### A.3  DATA

### A.3.1  MIMIC-III KNOWLEDGE GRAPH PROCESSING

We utilize the publicly available MIMIC-III clinical database Johnson et al. (2016), which comprises de-identified health records from over 40,000 ICU admissions between 2001 and 2012. The

| patient_id | visit | diagnoses | procedures | medications | proteins |
|---|---|---|---|---|---|
| 0 | 0 | 1 | ['4239', '5119', '78551', '4589', '311', '7220... | ['3731', '8872', '3893'] | ['N02B', 'A01A', 'A02B', 'A06A', 'B05C', 'A12A... | [PROTEIN:3897, PROTEIN:6839] |
| 1 | 0 | 2 | ['7455', '45829', 'V1259', '2724'] | ['3571', '3961', '8872'] | ['N02B', 'A01A', 'A02B', 'A06A', 'A12A', 'B05C... | [PROTEIN:3897, PROTEIN:6839] |
| 2 | 1 | 1 | ['41071', '78551', '5781', '5849', '40391', '4... | ['0066', '3761', '3950', '3606', '0042', '0047... | ['A06A', 'B05C', 'C07A', 'A12B', 'C03C', 'A12A... | [PROTEIN:1621, PROTEIN:4405, PROTEIN:4876, PRO... |
| 3 | 2 | 1 | ['2252', '3485', '78039', '4241', '4019', '272... | ['0151'] | ['B05C', 'A07A', 'C07A', 'A06A', 'N02B', 'C02D... | [PROTEIN:3897, PROTEIN:4876, PROTEIN:11410, PR... |
| 4 | 2 | 2 | ['41401', '4111', '4241', 'V4582', '2724', '40... | ['3613', '3615', '3961', '8872', '9904', '9905... | ['N02B', 'A01A', 'A02B', 'A06A', 'A12A', 'B05C... | [PROTEIN:3897, PROTEIN:6839] |

Figure 5: An illustrative subset of the cleaned MIMIC-III dataset showing multi-modal alignment of diagnoses, medications, and procedure-derived symptoms for individual patient visits.

dataset includes rich, time-stamped information across multiple domains, including medications, diagnoses, procedures, demographics, and vital signs, making it a robust foundation for modeling patient trajectories and building clinical knowledge graphs.

To convert raw electronic health records (EHRs) into a structured knowledge graph, we extend the preprocessing pipeline introduced in SafeDrug Yang et al. (2021). Our aim is to ensure both temporal consistency and semantic completeness across the three principal clinical modalities: medications, diagnoses, and procedures.

We begin by discarding records lacking valid timestamps or patient identifiers. Only encounters with temporally aligned entries across all three modalities are retained. Medication records are cleaned by removing non-essential details such as dosages, administration routes, and free-text annotations. To reduce data sparsity while preserving clinical diversity, we limit diagnostic and procedural codes to the most frequently occurring entries. Temporal normalization is applied to align visit dates, and missing data is handled via forward imputation within individual patient timelines. Patients with a single visit or incomplete records are excluded to ensure sufficient longitudinal depth.

For improved semantic interpretability, raw clinical codes are mapped to standardized vocabularies. Medications are categorized into higher-level therapeutic groups using the Anatomical Therapeutic Chemical (ATC) classification system. Procedural codes are translated into symptom-related descriptors based on a curated procedure-symptom ontology. In cases where no mapping is available, placeholder tokens are inserted to preserve sequence continuity.

Finally, we merge the cleaned data across all modalities using shared patient and visit identifiers. The resulting dataset comprises multi-modal clinical snapshots, where each visit encodes the patient's diagnostic history, prescribed treatments, and presenting symptoms—providing a rich foundation for graph-based modeling and analysis. An illustrative subset of the cleaned and structured cohort is visualized in Figure 5.

### A.3.2    OREGANO KNOWLEDGE GRAPH

We construct the Oregano Knowledge Graph by following the official pipeline provided in the Awesome Biomedical Knowledge Graphs project (Haas et al., 2023)[2]. This pipeline integrates multiple biomedical ontologies and curated databases to generate a heterogeneous graph that supports reasoning over clinical and molecular concepts.

The graph incorporates diverse sources such as UMLS (Unified Medical Language System), DrugBank, MeSH, and SNOMED CT. Entities in the graph represent biomedical concepts—including diseases, drugs, procedures, and symptoms—while edges represent semantically typed relations such as `treats`, `causes`, `has_symptom`, and `interacts_with`.

To generate the graph, we clone the public repository and run the extraction and normalization scripts. Entity alignment is performed using UMLS Concept Unique Identifiers (CUIs) to ensure semantic consistency across vocabularies. The final graph is serialized in structured formats (e.g., CSV triples or RDF), with each triplet storing a `head` entity, `relation` type, and `tail` entity.

---

[2]Available at: Robert Haas et al., Awesome Biomedical Knowledge Graphs, 2023.

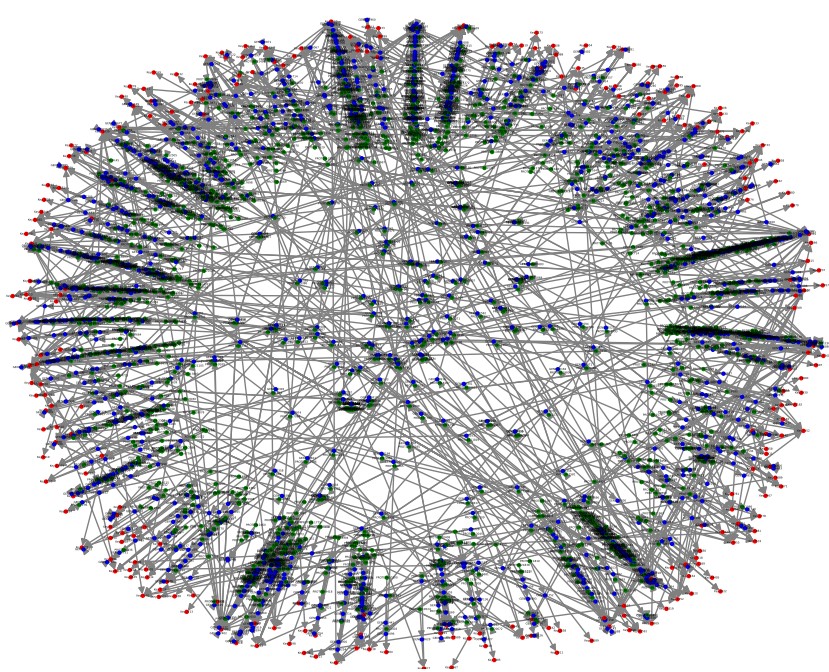

Figure 6: Overview of the Oregano Biomedical Knowledge Graph composed of curated entities and relations across clinical ontologies.

Figure 6 presents an overview of the Oregano Knowledge Graph, which serves as a core biomedical knowledge base for downstream tasks such as diagnostic reasoning, phenotype prediction, and drug repurposing.

## A.4 KNOWLEDGE REASONING RULE

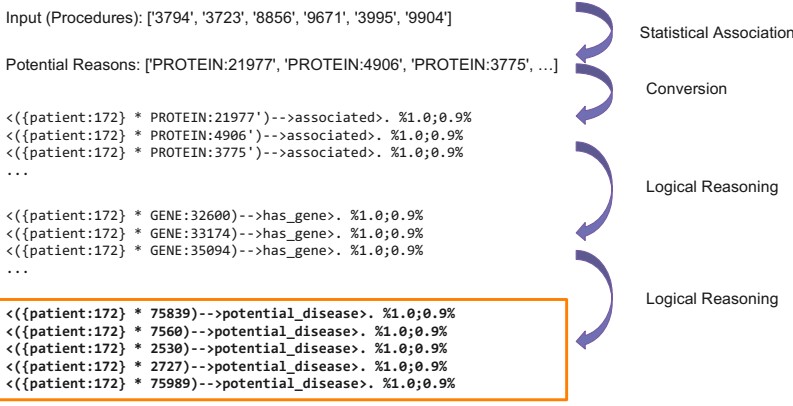

Figure 7: Illustration of the diagnostic reasoning process over the Oregano Knowledge Graph. The input is associated with "potential reasons" and converted to *Narsese* knowledge representations. The reasoning engine executes two steps of logical reasoning to derive new knowledge, indicating the potential disease of a patient (bold).

## A.5 HYPERPARAMETER SETTINGS OF KNOWLEDGE ALIGNMENT

All experiments are conducted in Python. The Word2Vec model is implemented using the `gensim` library, while classification and evaluation procedures are carried out with `scikit-learn`. This

Figure 8: ReCLLaMA user interface for diagnostic question answering, enabling structured outputs and evidence-grounded explanations.

pipeline provides a reproducible and interpretable foundation for aligning clinical and biomedical knowledge sources. The complete configuration of the experimental setup is as follows:

- **Embedding dimension:** 100
- **Context window size:** 5
- **Word2Vec model:** Skip-gram (`sg=1`)
- **Negative sampling ratio:** 1:1 (balanced)
- **Random Forest:** 100 estimators, maximum depth of 10
- **Train/test split:** 80% / 20% (stratified)
- **Random seed:** 42

## A.6 RESULTS

### A.6.1 RECLLAMA INTERFACE OVERVIEW

We present the user interface of the ReCLLaMA system in Figure 8. The interface is designed to support real-time, multi-turn interactions with users via natural language. It allows patients or clinicians to pose clinical questions and receive structured, interpretable diagnostic feedback. The system maintains session state across turns, enabling context-aware dialogue and follow-up queries (e.g., "Explain how Gene 4254 affects heart function"). Supporting evidence—such as gene-disease associations, protein annotations, and mechanistic descriptions—is embedded in expandable sections for traceability and interpretability.

### A.6.2 DIAGNOSTIC RESPONSE COMPARISON

To evaluate ReCLLaMA's diagnostic reasoning capabilities, we conduct a comparative study using a real-world test case introduced in the last part of Result section. We benchmark the system against ground truth expert annotations and ChatGPT-4.0's generative outputs. Figures 9, 10, and 11 provide visual comparisons of diagnostic explanations generated by each source:

- Figure 9 shows the reference explanation curated by human medical experts.

- Unspecified septicemia
- Pneumonitis due to inhalation of food or vomitus
- Acute on chronic systolic heart failure
- Congestive heart failure, unspecified
- Acute respiratory failure
- Subendocardial infarction, initial episode
- Atrial fibrillation
- Severe sepsis
- Diabetes mellitus without mention of complication, type II or unspecified type, not stated as uncontrolled
- Other persistent mental disorders due to conditions classified elsewhere
- Pure hypercholesterolemia
- Old myocardial infarction
- Esophageal reflux
- Long-term (current) use of anticoagulants
- Unspecified essential hypertension
- Fitting and adjustment of artificial leg
- Fitting and adjustment of other specified prosthetic device

Figure 9: Ground-truth diagnostic explanation manually authored by clinical experts, serving as the gold reference.

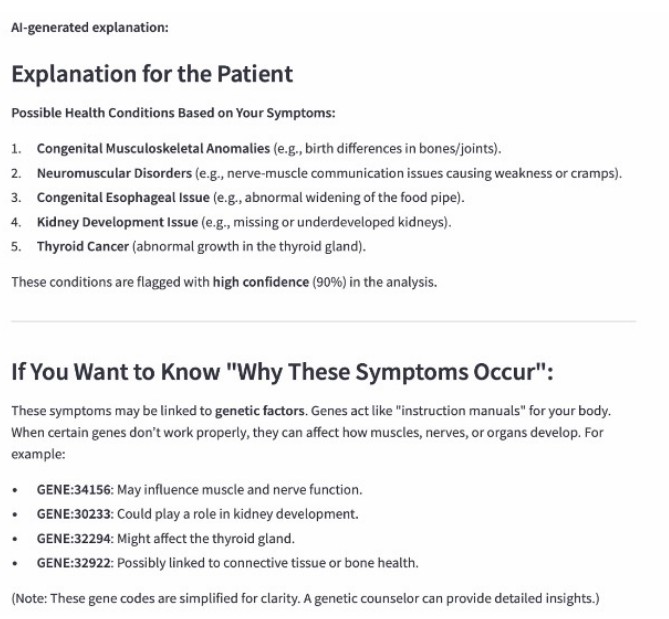

Figure 10: ReCLLaMA-generated explanation using symbolic biomedical reasoning over structured knowledge graphs.

- Figure 10 displays ReCLLaMA's generated explanation, grounded in symbolic reasoning and supported by biomedical knowledge graphs.
- Figure 11 illustrates the generative output from ChatGPT-4.0 for the same input query.

The comparison reveals that ReCLLaMA produces responses that are more aligned with expert knowledge and traceable to biomedical sources, while ChatGPT-4.0 provides fluent but less grounded outputs.

Based on your detailed symptoms, here's an AI-generated medical explanation integrating potential diagnoses and related biological mechanisms:

**Likely Diagnoses (based on ICD-9 symptom codes and clinical interpretation):**

1. **Photosensitivity Reaction**
   - **ICD-9:** 692.72 — Contact dermatitis due to solar radiation
   - Your description of severe skin blistering and redness upon sunlight exposure is characteristic of photosensitivity, which can be **drug-induced** (especially antibiotics or sedatives).

2. **Drug-Induced Toxicity / Overdose Syndrome**
   - **ICD-9:** 969.4 — Poisoning by benzodiazepines and other sedative-hypnotics
   - Sedation, confusion, muscle weakness, and cardiovascular effects like racing/fluttering heart suggest CNS depressant effects.

3. **Gastrointestinal Distress / Acute Gastritis**
   - **ICD-9:** 535.4 — Other specified gastritis with hemorrhage
   - Severe cramping, vomiting bile, and doubled-over pain point to acute irritation of the gastrointestinal lining.

4. **Autonomic Dysregulation / Panic-like Toxic Response**
   - **ICD-9:** 780.2 — Syncope and collapse
   - The sensation of gasping for air, rapid pulse, and blurred vision may result from systemic

Figure 11: ChatGPT-4.0 response generated without access to structured biomedical knowledge or symbolic inference mechanisms.

### A.6.3 SYSTEM DEMONSTRATION

To complement the screenshots, we also provide a video demo that illustrates ReCLLaMA's real-time interaction on the same test case. The demo showcases the system's ability to extract clinical entities, perform inference over the knowledge graph, and generate structured diagnostic responses with supporting evidence. The demonstration is available as supplementary material.