# OpenReview forum: "ReCLLaMA: A Reasoning-Centered LLM Agent for Medical Diagnosis"
_ICLR.cc/2026/Conference — ICLR 2026 Conference Withdrawn Submission_

### Official Review · Reviewer_yuwe · 2025-10-29

**Soundness:** 3
**Presentation:** 3
**Contribution:** 1
**Rating:** 2
**Confidence:** 3

**Summary:**

This paper propose ReCLLaMA, which is a clinical agent designed to improve medical diagnosis by combining LLMs with symbolic reasoning. It first uses an LLM to understand a free-text medical descriptions and convert them into standardized medical elements. Then, its core innovation lies in using a symbolic reasoning engine (NARS) over biomedical knowledge graphs to logically infer diagnoses. This hybrid approach aims to provide accurate, interpretable, and evidence-based diagnostic hypotheses.

**Strengths:**

1. The paper is easy to follow, with the components of the agent pipeline clearly described. This modular breakdown enhances reproducibility and understanding of the system's workflow.

2. The work addresses a critically important task: providing interpretable medical reasoning and diagnosis. This focus on explainability is essential for building trust and facilitating the adoption of AI in clinical decision-making.

**Weaknesses:**

1. Lack of Novelty. The core premise of the paper is to combine LLMs with symbolic reasoning over knowledge graphs (KGs). However, the architectural design assembles existing components, and demonstrates a lack of innovation.

- LLM as a Mere Front-End: The LLM is used as a simple pre-processing role, i.e., parsing natural language queries. This usage of LLMs as an "interface" to structured databases is now a standard and well-established practice and does not constitute a novel contribution.

- Conventional Knowledge Graph Reasoning: The "reasoning" part of the system is handled by a traditional symbolic reasoning engine (i.e., the Non-Axiomatic Reasoning System, NARS 2013) operating on a pre-existing knowledge graph. The NARS is based on work from 2013, and KG-based reasoning for medical diagnosis is a mature field with extensive prior works.

- A "Pipelined" Assembly, rather than an Integrated Solution: The work essentially chains together an LLM front-end with an reasoning backend. The primary contribution appears to be the engineering of this pipeline. The combination itself does not seem to produce emergent capabilities that neither component could achieve independently. I would expect more innovative integration than improves the reasoning capability of LLMs with KG.

2. About the Experimental Design and Results
- Insufficient Test Set Size: The experiments are conducts on mimic 3, while mimic 4 is released. Furthermore, the key diagnostic accuracy are conducting on only 100 held-out cases (e.g., Table 2). Such a small sample size cannot reliably measure performance, generalize findings, or detect improvements over baselines.

- Unexplained Data Selection: The paper fails to detail how these 100 test cases were selected.

- Low Performance Metrics: The results reported in Table 2 are low. Key metrics like Precision, Recall, and F1-score are near zero (0.0100, 0.0033, and 0.0050, respectively). Furthermore, several ablated model variants in Table 3 report scores of 0.0000. This may indicate that the system performs very poorly.

- Weakened Baselines: The choice of baselines (e.g., "MDAgent," "KG-CoT") are not represent the state-of-the-art. A more compelling comparison would be against more strong baselines, fine-tuned LLM or other well-established clinical decision support system.

**Questions:**

Please refer to the questions raised in the "Experimental Design" part in the weakness.

---

### Official Review · Reviewer_cEkJ · 2025-10-31

**Soundness:** 3
**Presentation:** 2
**Contribution:** 3
**Rating:** 2
**Confidence:** 3

**Summary:**

The submission introduces ReCLLaMA, a modular, Reasoning-Centered LLM Agent designed for interpretable and accurate medical diagnosis by explicitly integrating subsymbolic and symbolic AI techniques. This agent processes free-text symptom queries using a BioBERT-LLM hybrid for entity extraction, followed by a statistical alignment step using Random Forest over embedded entities to unify heterogeneous biomedical knowledge graphs (EHR and Drug KGs). Central to the framework is a symbolic reasoning engine built upon Non-Axiomatic Logic (NAL), which performs traceable deductive and abductive inference under uncertainty to derive ICD-9 diagnostic hypotheses, culminating in LLM-generated, patient-friendly explanations grounded in KG traces. Empirical results on MIMIC-III and Oregano KG data demonstrate that the system achieves high diagnostic accuracy and calibrated confidence superior to comparable LLM and KG-CoT baselines.

**Strengths:**

1. The framework offers an original, theoretically grounded integration of subsymbolic LLMs and symbolic Non-Axiomatic Logic for clinical decision support.
2. Modularity in the design, encompassing extraction, alignment, reasoning, and translation, enhances both clarity and engineering robustness.
3. The reasoning component successfully mitigates LLM hallucination by explicitly grounding diagnostic pathways in structured knowledge graph traces.
4. Empirical evaluation demonstrates superior performance in structured diagnostic accuracy and confidence calibration compared to strong baselines.

**Weaknesses:**

1. The statistical knowledge alignment backbone, relying on Word2Vec embeddings and Random Forest, appears relatively simple compared to modern Graph Neural Network alignment methods.
2. The claimed full integration of the abductive, deductive, and inductive cycle lacks explicit demonstration of the iterative rule discovery or hypothesis revision stage.
3. Quantitative results demonstrating diagnostic superiority are based on a relatively small cohort of 100 test cases, potentially limiting the statistical significance of the claims.
4. The reasoning system's reliance on fixed background rules K1 and K2 suggests limited ability for autonomous knowledge discovery or adaptation within the NAL framework.
5. Detailed results for the knowledge alignment module are reported on a synthetic pairwise symptom-protein dataset which may not accurately reflect real-world KG linking difficulty.
6. Specific ICD-9 codes and the choice of the Oregano KG may constrain the immediate applicability and update burden of the system in evolving clinical environments.
7. The ablation study (Table 3) shows that removing the Random Forest alignment backbone drastically reduces performance to zero, suggesting this statistical component may be overly relied upon.

**Questions:**

1. Authors should clarify exactly how the framework implements the inductive phase of the Peirceian reasoning cycle, specifically addressing how rules (like K1 or K2) are actually discovered or revised based on observations.
2. Could the authors provide latency benchmarks for the Module IV symbolic reasoning component, especially for complex multi-step deductions, and discuss its scalability for real-time deployment?
3. Beyond the concatenated Word2Vec embeddings, please list and discuss any other features used by the Random Forest classifier in Module III to predict the validity of biomedical relations.

---

### Official Review · Reviewer_xHmv · 2025-11-03

**Soundness:** 1
**Presentation:** 1
**Contribution:** 1
**Rating:** 2
**Confidence:** 4

**Summary:**

This paper introduces ReCLLaMA, a "Reasoning-Centered LLM Agent for Medical Diagnosis." The authors aim to address the limitations of Large Language Models (LLMs) in clinical diagnosis, specifically their tendency for hallucination and lack of interpretable reasoning. The proposed framework consist of 5 modules: user interface, knowledge extraction, knowledge alignment, knowledge reasoning, and knowledge translation. The authors conduct experiments on each module to evaluate the effectiveness of the proposed method.

**Strengths:**

1. The paper is easy to follow and understand.
2. The paper attempts to solve the correct and critical problems in medical AI: hallucination, lack of interpretability, and the need for formal, grounded reasoning.
3. The 5-module design (Extraction, Alignment, Reasoning, etc.) is a logical way to separate concerns.

**Weaknesses:**

1. The complete evaluation system in the paper is very confusing. There are no detailed descriptions of how the evaluations were done and what the meanings of the evaluation results are. The results in the Table 2 and 3 are basically all 0 or near-zero. I don't see any reliable results can be drawn from the results like these. And the same for the results in table 4, where all numbers are almost 100 for both the proposed method and compared baselines.
2. The use of Word2Vec and Random Forest for a critical alignment task is a weak methodological choice, where these 2 methods are very out-dated and there are no clear indications of why no more recent methods are used or compared.
3. The "reasoning-centered" part of the agent relies entirely on applying NARS, an existing external logic system. The novelty of the framework is very limited.

**Questions:**

1. How can the paper claim 92.8% accuracy for extraction, ~99% for alignment, and 81% for reasoning, yet the final end-to-end system in Table 2 achieves only 5% "Any-Hit" accuracy? This implies a catastrophic failure at the interfaces between modules. Where exactly is the system breaking down?
2. he ablation in Table 3 shows that removing any component (CE, RF, or Reasoner) results in a score of 0.0000 across all metrics. This seems statistically extreme. Does this mean not a single test case could be correctly processed if even one module was changed? Can you explain this total pipeline collapse?

---

### Note · Authors · 2025-11-12

I have read and agree with the venue's withdrawal policy on behalf of myself and my co-authors.